# Two Types of Liposomal Formulations Improve the Therapeutic Ratio of Prednisolone Phosphate in a Zebrafish Model for Inflammation

**DOI:** 10.3390/cells11040671

**Published:** 2022-02-15

**Authors:** Yufei Xie, Panagiota Papadopoulou, Björn de Wit, Jan C. d’Engelbronner, Patrick van Hage, Alexander Kros, Marcel J. M. Schaaf

**Affiliations:** 1Institute of Biology, Leiden University, 2333 CC Leiden, The Netherlands; xie_yufei1990@hotmail.com (Y.X.); b.de.wit.3@umail.leidenuniv.nl (B.d.W.); jan.dengelbronner@xs4all.nl (J.C.d.); p.van.hage@umail.leidenuniv.nl (P.v.H.); 2Institute of Chemistry, Leiden University, 2333 CC Leiden, The Netherlands; p.papadopoulou@lic.leidenuniv.nl (P.P.); a.kros@chem.leidenuniv.nl (A.K.)

**Keywords:** glucocorticoids, inflammation, zebrafish, laser wounding, liposome

## Abstract

Glucocorticoids (GCs) are effective anti-inflammatory drugs, but their clinical use is limited by their side effects. Using liposomes to target GCs to inflammatory sites is a promising approach to improve their therapeutic ratio. We used zebrafish embryos to visualize the biodistribution of liposomes and to determine the anti-inflammatory and adverse effects of the GC prednisolone phosphate (PLP) encapsulated in these liposomes. Our results showed that PEGylated liposomes remained in circulation for long periods of time, whereas a novel type of liposomes (which we named AmbiMACs) selectively targeted macrophages. Upon laser wounding of the tail, both types of liposomes were shown to accumulate near the wounding site. Encapsulation of PLP in the PEGylated liposomes and AmbiMACs increased its potency to inhibit the inflammatory response. However, encapsulation of PLP in either type of liposome reduced its inhibitory effect on tissue regeneration, and encapsulation in PEGylated liposomes attenuated the activation of glucocorticoid-responsive gene expression throughout the body. Thus, by exploiting the unique possibilities of the zebrafish animal model to study the biodistribution as well as the anti-inflammatory and adverse effects of liposomal formulations of PLP, we showed that PEGylated liposomes and AmbiMACs increase the therapeutic ratio of this GC drug.

## 1. Introduction

Glucocorticoids (GCs) are a class of steroid hormones secreted by the adrenal gland. The main endogenous GC in our body is cortisol, which has been shown to have potent anti-inflammatory effects. Synthetic analogs of cortisol, such as prednisolone and dexamethasone, are therefore widely prescribed for treating a variety of immune-related diseases, e.g., asthma, rheumatoid arthritis (RA), dermatitis, and several autoimmune diseases [1,2]. Through activation of an intracellular receptor, the glucocorticoid receptor (GR), GCs regulate a wide variety of systems in our body, including the immune system, metabolism, bone formation, and central nervous system, leading to side effects, such as infectious diseases, diabetes, osteoporosis, and depression, which severely limit the clinical use of these widely prescribed anti-inflammatory drugs [3]. Using local administration approaches, such as intra-articular injection, inhalation, and topical treatment, these side effects could be alleviated to some extent, but these methods are applicable for only a limited number of diseases [4]. 

Using nanoparticles to achieve targeted drug delivery is a promising approach to increase the specificity of a drug and to reduce its side effects [5,6]. There are two major types of targeted delivery approaches: active and passive targeting. Active targeting is generally achieved by modifying the surface of particles with specific targeting molecules, such as antibodies, peptides, or carbohydrates, which may be recognized by antigens or other molecules on the membranes of specific cells so the drugs can be released at a desired location [6]. The surface of passive targeting nanoparticles is generally not altered with targeting molecules. The targeting could be dependent on the enhanced permeation and retention (EPR) effect which is observed in tumors and inflammatory sites due to the locally increased permeability of the vasculature [7]. In addition, recent studies on the in vivo interactions of nanoparticles have demonstrated that the adherence of serum proteins, such as complement proteins, immunoglobulins, and non-immune opsonins, causes opsonization and alters the biodistribution of nanoparticles through enhancing their clearance by specialized cells of the reticuloendothelial system (RES), which in mammals are mainly located in the liver and spleen [8,9]. Since the efficacy and specificity of this type of targeting may be influenced by several physicochemical properties of the nanoparticles, including the size, shape, surface charge, composition and surface modification, it provides a framework for the rational design of drug delivery systems based on nanoparticle engineering [10,11,12].

Liposomes are spherical lipid-based nanoparticles that have widely been investigated for delivering therapeutics. The composition of liposomes can easily be manipulated, which alters their physicochemical properties, leading to a plethora of formulations [13,14]. They represent the most successful type of nanoparticles, with a great number of FDA-approved formulations (e.g., AmBisome, Myocet, and Doxil) [6,15]. One frequently studied formulation of neutrally charged liposomes uses phospholipids linked to a polymer polyethylene glycol (PEG) chain [16,17]. The PEGylation prevents the liposome from being recognized by serum proteins and thereby prevents opsonization and clearance, which increases its half-life in the circulation [18].

Significant efforts have been made to explore the use of long-circulating liposomes for targeted delivery of GCs. A widely investigated formulation consists of the lipids PEG2000-distearoyl phosphatidylethanolamine (DSPE), dipalmitoyl phosphatidylcholine (DPPC) and cholesterol, and the water-soluble GC prednisolone phosphate (PLP) as the loaded drug [19,20]. In a rodent model of RA, the PEGylated liposome-encapsulated PLP remained in the circulation longer than free PLP and accumulated in the inflamed joints, resulting in enhanced therapeutic effects compared to free PLP [19,21]. Similar results were obtained using other rodent models for inflammation-related diseases such as atherosclerosis [22], multiple sclerosis [16], and cancer [23]. Attempts to further improve the therapeutic effects include using different synthetic GCs [24,25,26], conjugation with targeting molecules [27,28,29,30], optimization of formulation [31,32], and co-treatment with anti-tumor drugs [33]. However, obstacles still exist such as unwanted off-target accumulation in spleen, liver, and kidney [21,34]; the level of GC-induced side effects [25,26]; an unexpected lack of anti-inflammatory effect when applied clinically [35]. Therefore, novel approaches should be investigated, such as targeted delivery of GCs to macrophages [36,37,38,39,40,41], which are critical components of the inflammatory response and, therefore, represent one of the target cells for the treatment of inflammatory diseases. A previous study in which macrophage-targeted GC delivery was applied in a mouse multiple sclerosis model showed full therapeutic efficacy of the drug [42]. In addition, novel tools are needed, which can be used for rapid screening of the biodistribution, therapeutic effects, and side effects of different liposomal formulations of GC drugs. 

Over the last decades, the zebrafish has emerged as a useful in vivo model for biomedical research [43], complementary to rodent models due to the fact of several characteristics such as their short generation time and the small size and optical transparency during the embryonic stage. In addition, the sequencing of the zebrafish genome [44], the availability of various genetic tools [45] and their well-conserved immune system [46] have contributed to the versatility of this animal model. Due to the similarity of the GC signaling pathway between humans and zebrafish, the zebrafish model is highly suitable for studying the effects and mechanisms of GC action in vivo, particularly their anti-inflammatory effects [47,48,49,50]. 

In several recent studies, the zebrafish has been employed to investigate drug delivery using different liposomal formulations. The transparency of the embryos allows for the direct visualization of the biodistribution of fluorescently labeled liposomes including their presence in the circulation, interaction with specific cell types, and uptake and accumulation in tissues [51,52,53,54,55]. This makes the zebrafish an excellent tool for pre-screening of liposomal drug formulations. Using the zebrafish model, it was demonstrated that neutrally charged liposomes of ~100 nm in size freely circulate in the vasculature of the embryos. In contrast, anionic liposomes were found to strongly interact with cells from the RES, in particular scavenging endothelial cells (SECs, via a stabilin-mediated clearance pathway) and blood-resident macrophages, and cationic liposomes were shown to non-specifically interact with anionic surfaces of all cells (especially at higher surface charges), including blood vessel walls, in addition to clearance by the RES [51,52,53,54].

In the present study, we used the zebrafish embryo as a model system to both visualize the biodistribution of liposomal formulations incorporating PLP and determine the therapeutic ratio of the encapsulated drug. Using this model, two formulations which differ in composition and targeting design, were investigated: long-circulating PEGylated liposomes and novel macrophage-targeting liposomes (AmbiMACs). We demonstrate that both types of liposomes accumulate at sites of inflammation and that the encapsulation of PLP in these liposomes enhances the anti-inflammatory effects of this GC drug, whereas it decreases possible adverse effects.

## 2. Materials and Methods

### 2.1. Zebrafish Lines and Maintenance

Zebrafish were maintained and handled according to the guidelines from the Zebrafish Model Organism Database (http://zfin.org, accessed on 30 January 2022) and in compliance with the directives of the local animal welfare committee of Leiden University. They were exposed to a 14 h light and 10 h dark cycle to maintain circadian rhythmicity. Fertilization was performed by natural spawning at the beginning of the light period. Eggs were collected and raised at 28 °C in egg water (60 µg/mL Instant Ocean sea salts and 0.0025% methylene blue). The following fish lines were used in this study: the wild-type (wt) strain AB/TL, the transgenic lines *Tg*(*mpeg1:GFP^gl22^*) [56], *Tg*(*mpx:GFP^i114^*) [57], and *Tg*(*9xGCRE-HSV.Ul23:EGFP ^ia20^*) [58].

### 2.2. Liposome Preparation and PLP Encapsulation

All liposomes were formulated by extrusion (mini extruder, Avanti Polar Lipids) at temperatures above the *Tm* of all lipids (i.e., 65–70 °C). The total lipid concentration of liposomes was 5 mM, unless stated otherwise. All formulations contained 1% mol of DOPE-Lissamine Rhodamine (DOPE-LR, excitation maximum at 560 nm and emission maximum at 583 nm) for visualization by fluorescence microscopy. Macrophage targeting liposomes were formulated in ddH_2_O and PEGylated liposomes (van der Valk et al., 2015) were formulated in phosphate buffered saline (PBS). Briefly, stock solutions (1–10 mM) of lipids in chloroform (for DSPG, 1 mM stock was prepared in chloroform:methanol in a 5:1 ratio) were mixed at the desired molar ratios and dried first under N_2_, then in vacuo for >1 h. The resulting lipid films were rehydrated with 1 mL aqueous solvent (ddH_2_O or PBS) at 65–70 °C, with gentle vortexing. Large unilamellar vesicles with a size ~100 nm were formed by passing the hydrated lipids 11 times through 2 × 400 nm polycarbonate (PC) membranes (Nucleopore Track-Etch membranes, Whatman), followed by 11 times through 2 × 100 nm PC membranes. All liposomes (with or without encapsulated PLP) were prepared freshly before injection.

Prednisolone disodium phosphate (PLP, MedChemExpress) was encapsulated by hydrating the lipid film with an aqueous solution of 50 mg/mL PLP. After extrusion the unencapsulated PLP was removed by size exclusion chromatography (NAP^TM^ 25 columns Sephadex^TM^, GE Healthcare) with elution solvent ddH_2_O or PBS. The encapsulated amount of compound was determined by the absorbance measured using UV spectrophotometry. For this purpose, a calibration curve of PLP dissolved in MeOH was made and found to be linear at concentrations of 1–40 μg/mL (Appendix A). The liposomal solution was diluted 20 times in MeOH and vortexed for membrane disruption and PLP release and the absorbance of PLP was subsequently measured at 242 nm. Reported amounts of PLP are total amounts in the solution. Injection dose was determined every time after liposome preparation and before administration.

Theoretical encapsulation efficiency (*EE%_theoretical_*) and measured encapsulation efficiency (*EE%_measured_*) were determined as follows:$$\begin{array}{c} EE{\%}_{measured}=\frac{C_{encapsulated}}{C_{initial}}\times 100\\ EE{\%}_{theoretical}=\frac{C_{expected}}{C_{initial}}\times 100=\frac{C_{initial}\;\times \;\Phi_{inner}}{C_{initial}}\times 100 \end{array}$$
where *C**_encapsulated_* is the PLP concentration after removal of the free drug (by SEC), as determined by the absorbance of PLP via UV–Vis and its calibration curve (Appendix A), multiplied by 20 (due to the dilution in MeOH for liposome disruption) and by 2.5 (due to the dilution during SEC); *C**_initial_* is the initial PLP concentration used to hydrate the lipid film before extrusion; *C**_expected_* is the theoretical PLP concentration that can be entrapped, based on the total inner volume of the liposomal core (*Φ_inner_*) [59,60] (Appendix A).

Macrophage-targeting liposomes (AmbiMACs) consisted of dioleoyl phosphatidyl choline (DOPC), distearoyl phosphatidyl glycerol (DSPG), and cholesterol with molar ratios of either 50:10:40, 50:15:35, 50:20:30, 50:25:35, or 50:30:20. The PEGylated liposomes consisted of dipalmitoyl phosphatidyl choline (DPPC), cholesterol, and distearoyl-sn-glycero-3-phosphoethanolamine-N-[amino(polyethylene glycol)-2000] (PEG-DSPE) with a molar ratio of 62:5:33.

### 2.3. Cryogenic Transmission Electron Microscopy for Liposome Imaging

Liposomes (3 μL, 5 mM total lipid concentration) were applied to a freshly glow-discharged carbon 200 mesh Cu grid (Lacey carbon film, Electron Microscopy Sciences, Aurion, Wageningen, The Netherlands). Grids were blotted for 3 s at 99% humidity in a Vitrobot plunge-freezer (FEI VitrobotTM Mark III, Thermo Fisher Scientific, Waltham, MA, USA). Cryo-EM images were collected on a Talos L120C (NeCEN, Leiden University) operating at 120 kV. Images were recorded manually at a nominal magnification of 36,000× yielding a pixel size at a specimen of 2.9 ångström (Å).

### 2.4. Injection of Drugs

All treatments were given intravenously to the zebrafish embryos. After anesthesia with 0.02% aminobenzoic acid ethyl ester (tricaine, Sigma–Aldrich, St. Louis, MO, USA), 2 or 3 dpf embryos were injected with control solution (water), empty liposomes, different amounts of free PLP, or liposome-encapsulated PLP in the duct of Cuvier under a Leica M165C stereomicroscope. For free PLP, the injected volume was 1 nl and the concentration of the injected PLP solution (in ddH_2_O) varied from 25 mM PLP to 0.04 mM to achieve an injected amount of 25–0.04 pmol. For liposome-encapsulated PLP, the injected amount of 1 pmol was achieved by injecting the original liposome solution with a volume calculated based on the concentration determined by UV spectrophotometry. Lower doses were injected using the same volume and a lower concentration of liposomes. As a control for testing the vascular permeability, 1 nL of 20 μg/mL dextran was injected (fluorescently labeled with tetramethylrhodamine, 2,000,000 MW, Invitrogen, Waltham, MA, USA).

### 2.5. Microscopy

For confocal laser scanning microscopy, anesthetized embryos were mounted in 1% low-melting agarose in egg water containing 0.02% tricaine on 40 mm glass-bottom dishes (WillCo-dish, WillCo Wells, Amsterdam, The Netherlands). Images were taken using a Leica TCS SP8 confocal microscope with a 10× (NA 0.4) or 20× (NA 0.75) objective and a 488 and/or 532 nm excitation laser light (Figure 1, Figure 2 and Figure 3 and Appendix A). For other brightfield and/or fluorescence microscopy imaging, anesthetized embryos were imaged using a Leica M205FA fluorescence stereomicroscope equipped with a Leica DFC 345FX camera (Figure 4, Figure 5 and Figure 6).

### 2.6. Laser Wounding

Adapted from the method of yolk wounding using laser irradiation in zebrafish embryos described previously [61], we used laser wounding to a region in the tail, since this area is vascularized, and the thin tissue allows convenient imaging of accumulated leukocytes. Anesthetized 3 dpf embryos were mounted in 1% low-melting agarose in egg water containing 0.02% tricaine on a microscope slide (VWR). A 100 μm long burning wound was created in the tail by laser irradiation with a ZEISS PALM Microbeam Laser Microdissection system using a 20× objective (NA 0.4) (Figure 3A). Drug treatment by intravenous injection was performed immediately after wounding. The number of recruited neutrophils was determined at 4 h after wounding.

### 2.7. Regeneration Assay

The tails of anesthetized 2 dpf embryos were partially amputated with a 1 mm sapphire blade (World Precision Instruments, Sarasota, FL, USA) on 2% agarose-coated Petri dishes under a Leica M165C stereomicroscope (Figure 5A). Drug treatment by intravenous injection was performed immediately after amputation. The length of fin regeneration was determined from microscopic images taken at 36 hpa as previously described [62].

### 2.8. Image Analysis

To determine the (co)localization of liposomes and macrophages in the zebrafish embryos, macrophages were detected based on their fluorescent GFP label and liposomes on their fluorescent DOPE-LR label. The number of macrophages containing liposomes (Figure 1G and Figure 2I) were counted manually in the whole embryo based on the colocalization of the GFP and DOPE-LR signals (the z-stacks were examined to help determine this colocalization). To subsequently determine the percentages of macrophages that contained liposomes (Figure 1H and Figure 2J), the total number of macrophages was determined in the whole fish based on the GFP signal of these cells. To quantify the ratio of the liposome signal between the CHT/CV area and the dorsal part of the tail (Figure 1I and Figure 2K), a maximum intensity projection of the z-stacks was generated and, subsequently, the intensity of the liposome signal in the indicated areas was determined using the Analyze and Measure tool in the ImageJ software. To determine neutrophil migration upon laser wounding, neutrophils were detected based on their fluorescent GFP label, and to quantify the number of recruited neutrophils in the fluorescence microscopy images of the wounded tails, the cells in a defined area of the tail were counted manually (Figure 4A). For the quantitation of the regeneration assay, in the images of the tail fins the length of the regenerated tissue was determined from the center of the original plane of amputation to the tip of the regenerating fin (Figure 5). Quantitation of transactivation of a GRE-containing promoter in the *Tg(9xGCRE-HSV.Ul23:EGFP ^ia20^)* was conducted on images taken at 24 hpi, by determining the relative EGFP signal in the embryonic body using the Analyze and Measure tool in the ImageJ software and determining the value “RawIntDen” (the sum of the values of the pixels in the image) (Figure 6A).

### 2.9. Statistical Analysis

Statistical analysis was performed using GraphPad Prism 7 by one-way ANOVA with Bonferroni’s post hoc test (Figure 1, Figure 2, Figure 4, Figure 5 and Figure 6) or two-tailed *t*-test (Figure 3F). Significance was accepted at *p* < 0.05.

## 3. Results

### 3.1. Macrophage-Targeting Liposomes (AmbiMACs) and PEGylated Liposomes Showed Different Biodistributions in Zebrafish Embryos

A new liposomal formulation, which has been shown to target macrophages and was named AmbiMAC, was used in the present study. This formulation originates from the marketed liposomal product AmBisome, which is composed of the anionic phospholipid 1,2-distearoyl-*sn*-glycero-3-phospho-*rac*-(1-glycerol) (DSPG), 1,2-distearoyl-*sn*- glycero-3-phosphocholine (DSPC), and cholesterol (ratio 21:53:26), and it is used for the liposomal formulation of the anti-fungal drug amphotericin B [63,64]. In a previous study, it was shown that upon intravenous administration of AmBisome in zebrafish embryos, these anionic liposomes interacted strongly with RES cell types, namely, scavenging endothelial cells (SECs, mediated by the scavenger receptor stabilin-2) and blood-resident macrophages [52]. In further studies, the formulation of AmBisome was altered by replacing the saturated phospholipid DSPC in the original formulation with the unsaturated 1,2-dioleoyl-*sn*-glycero-3-phosphocholine (DOPC). Interestingly, this newly designed AmbiMAC formulation not only interacted with blood-resident macrophages of the embryos but was also found to be associated with tissue-resident macrophages, most likely due to the altered rigidity of the liposomes, which may facilitate tissue penetration (unpublished data).

In the present study, we first altered the liposomal formulation of AmbiMACs to optimize the macrophage-targeting properties. For this purpose, fluorescently labeled liposomes were prepared with different DSPG:cholesterol ratios (i.e., 10:40%, 15:35%, 20:30%, 25:25%, and 30:20%), combined with 50% of DOPC. AmbiMACs with different DOPC:DSPG:cholesterol ratios were injected intravenously in 2 days post-fertilization (dpf) zebrafish embryos from the *Tg(mpeg1:GFP)* line, in which the macrophages are fluorescently labeled. At 2 h post-injection (hpi), confocal microscopy images were taken to study the biodistribution of the liposomes, and representative images are shown in Figure 1A–F. Using the images, we quantitated the number and percentage of macrophages that contained liposomes. The results show that a higher percentage of DSPG in the liposome formulation increases the macrophage-targeting efficiency of AmbiMACs (Figure 1G,H). Simultaneously, we observed association of AmbiMACs with endothelial cells (ECs) of the posterior cardinal vein (PCV), the caudal vein (CV), and the caudal hematopoietic tissue area (CHT). The uptake of liposomes by ECs in this region is considered as RES targeting, since the ECs in this area have been shown to be a functional equivalent of the liver sinusoidal/scavenger endothelial cells (LSECs) in mammals [52]. To quantitate this effect, we determined the ratio between the signal in the region encompassing the CHT and CV and the region dorsally from this area, which showed that increasing the DSPG percentage to 25% increased targeting of AmbiMACs to the region encompassing the CHT and CV (Figure 1I). Apparently, this RES targeting is altered upon modifying the physicochemical properties of the liposomes, most likely as a result of a difference in the rigidity of the particle, since the surface charge was only slightly altered by varying the DSPG percentage (Appendix A). Furthermore, the total number of macrophages in the embryonic body was not significantly influenced by the variation in DSPG percentages (Appendix A). After comparing the biodistribution of the different liposomes, AmbiMACs with 20% DSPG were considered as the optimal formulation, since they displayed the highest ratio between macrophage targeting and CHT/CV localization.

To characterize the behavior of the AmbiMACs (20% DSPG) in further detail, we injected this formulation in embryos at different ages (1, 2, and 3 dpf) and imaged their biodistribution at 2 hpi (Figure 2A–F). From the images, we quantified the number and percentage of macrophages that contained liposomes. The macrophage targeting property of AmbiMACs (20% DSPG) was observed at all ages. After injection at 1 and 2 dpf, around 65% of the macrophages had taken up liposomes (with a lower absolute number of macrophages at 1 dpf), and after injection at 3 dpf approximately 35% of all macrophages contained liposomes (Figure 2I,J and Appendix A). The targeting of AmbiMACs (20% DSPG) in the CHT and CV area, relative to the region dorsally from this area, was significantly higher at 1 dpf compared to 2 and 3 dpf (Figure 2K), likely due to the undeveloped vasculature in the dorsal region at 1 dpf. For comparison, a PEGylated liposomal formulation (DPPC(62%)/PEG-DSPE(5%)/cholesterol(33%)), which has regularly been used for encapsulation of PLP [19,20], was also injected in 3 dpf embryos. In contrast to AmbiMACs (20% DSPG), we observed that PEGylated liposomes were not taken up by macrophages (Figure 2G,H), which was reflected in a significantly lower percentage of the macrophages containing PEGylated liposomes (approximately 6%, Figure 2I,J). No difference was observed in targeting to the CHT and CV area between AmbiMACs (20% DSPG) and PEGylated liposomes at 3 dpf (Figure 2K).

Since we would like to use the AmbiMAC formulation for GC delivery to the inflamed tissues, we encapsulated the water-soluble GC PLP in AmbiMACs and in PEGylated liposomes (Appendix A). Encapsulation of PLP only resulted in minor changes in size and charge of the liposomes (Appendix A), and using cryo-electron microscopy, we showed that the morphology of the liposomes did not significantly change upon PLP encapsulation, although some bi- and multilamellar membranes were observed in the PLP-containing liposomes (Appendix A). Subsequently, the biodistribution of PLP-encapsulated liposomes was analyzed upon injection in zebrafish embryos, and the results showed that liposomes containing PLP behaved similarly to liposomes without PLP (Appendix A).

### 3.2. Liposomes Accumulated near the Wound after Laser Wounding

To study targeting of liposomes towards an inflammatory site, we damaged the tail of 3 dpf embryos from the *Tg(mpeg1:GFP)* line by laser irradiation (Figure 3A) and injected the two types of fluorescently labeled liposomes (AmbiMACs (20% DSPG) or PEGylated liposomes) immediately after the laser wounding. Confocal microscopy images were taken at 4 h post-wounding (hpw). In all images, accumulation of macrophages was observed near the wounded area (Figure 3B–E). In the embryos injected with the AmbiMACs (20% DSPG), most of these macrophages that had accumulated near the wounded area contained liposomes (Figure 3B,C). In the embryos injected with PEGylated liposomes, most of the macrophages near the wound site did not contain any liposomes, but accumulation of liposomes was observed in the wounded area with a diffuse pattern (Figure 3D). As a control for the integrity of the vascular system, the polysaccharide dextran (2,000,000 MW) was injected. No accumulation of dextran was observed near the wounded site (Figure 3E), suggesting that the accumulation of the PEGylated liposomes is not caused by local damage to the vascular system but is due to the inflammation-induced change in vesicle permeability. When we quantitated the percentages of macrophages containing liposomes in the area near the laser wound, we again found a very low percentage (3.2 ± 2.0%) in embryos injected with PEGylated liposomes, and a higher percentage (39.0 ± 5.2%) in the embryos injected with AmbiMACs (20% DSPG) (Figure 3F).

### 3.3. Encapsulated PLP Showed a More Potent Effect on Wounding-Induced Neutrophil Migration Than Free PLP

Since it has been shown in previous studies that macrophage migration towards a wounded site is not affected by GC treatment in zebrafish but that the neutrophil migration is inhibited [50,62], we used the accumulation of neutrophils at the wounded area as a readout for the anti-inflammatory effect of PLP. For this purpose, we applied laser wounding in 3 dpf embryos from the *Tg*(*mpx:GFP*) line, in which neutrophils are fluorescently labeled. At different time points after the wounding, fluorescence microscopy images were taken and the neutrophil accumulation in a defined area of the tail was quantitated (Figure 4A,B). The results showed that laser wounding induced a rapid increase in the number of accumulated neutrophils at early time points (from 0 to 4 hpw) and then the number decreased gradually between 4 and 24 after wounding (Appendix A). Subsequently, using the number of neutrophils at the site of laser injury at 4 hpw as a readout, we studied the anti-inflammatory effect of free PLP and liposome-encapsulated PLP. Injection of free PLP (i.e., 0.04, 0.2, 1, 5, and 25 pmol per embryo) resulted in a dose-dependent inhibition of the neutrophil migration, with a significant inhibitory effect observed for the 5 and 25 pmol doses (23.4 ± 3.6% and 30.9 ± 4.2% inhibition, respectively) (Figure 4C). Injection of PLP encapsulated in AmbiMACs (20% DSPG) (at doses of 0.04, 0.2, and 1 pmol) also inhibited neutrophil migration dose-dependently and showed a significant inhibition already at a dose of 0.2 pmol (21.0 ± 4.7%) (Figure 4D). Injection of PLP encapsulated in PEGylated liposomes resulted in a significant inhibition only at the 1 pmol dose (21.4 ± 6.0%) (Figure 4E). These results indicate that PLP shows a more potent anti-inflammatory effect when encapsulated in a liposome, with AmbiMACs (20% DSPG) showing a slightly higher potency than the PEGylated liposome.

### 3.4. Encapsulated PLP Had a Smaller Effect on Tissue Regeneration and Transactivation Than Free PLP

To develop novel anti-inflammatory GC therapies, it is important to study possible side effects of the treatment as well. In this study, we first investigated the effect of free PLP and encapsulated PLP on tissue regeneration. We performed tail fin amputation on 2 dpf embryos (Figure 5A) and injected the amputated embryos with different doses of free PLP or liposome-encapsulated PLP immediately after the amputation. The length of the regenerated tail fin was measured at 36 h post-amputation (hpa). Free PLP showed a significant inhibitory effect on regeneration at all doses tested (from 0.04 pmol (3.4 ± 1.0%) to 25 pmol (30.5 ± 3.1%)) and this effect was dose-dependent (Figure 5B,E). For PLP encapsulated in AmbiMACs (20% DSPG), a significant inhibition was observed when the injected dose was 1 pmol (5.7 ± 1.4%) but not at lower doses (0.04 pmol and 0.2 pmol) (Figure 5C). When the embryos were injected with PLP encapsulated in PEGylated liposomes, no significant effect of the treatment was observed (Figure 5D). These results indicate that encapsulation of PLP in liposomes decreases the inhibitory effect of PLP on regeneration, showing even an absence of inhibition for the PEGylated liposomes at the doses tested.

Subsequently, we studied the effect of free and liposome-encapsulated PLP on the transactivation of a glucocorticoid response element (GRE)-containing promoter. For this purpose, we used the *Tg(9xGCRE-HSV.Ul23:EGFP)* reporter line, in which the EGFP gene is driven by a GRE-containing promoter. This way, we were able to quantify the systemic induction of the transactivation properties of the Gr by the GC treatment, by quantitating the EGFP signal throughout the embryonic body. Embryos were injected with free PLP and liposome-encapsulated PLP at 3 dpf, and fluorescence microscopy images were taken at 24 hpi. The quantitated EGFP signals showed that both free and liposome-encapsulated PLP induced significant dose-dependent increases in the level of GFP expression (Figure 6). However, for free PLP a significant increase was already observed at a dose of 0.2 pmol (26.1 ± 2.6%), whereas for both AmbiMAC (20% DSPG) and the PEGylated liposome-encapsulated treatments, a significant increase was only observed for the 1 pmol dose (44.5 ± 3.4% and 6.2 ± 3.1% respectively). These data demonstrate that liposome encapsulation decreases the potency of PLP to induce the transactivation activity of the Gr throughout the body of zebrafish embryos.

Finally, we compared all observed effects of the 1 pmol dose of free PLP and PLP encapsulated in AmbiMACs (20% DSPG) and PEGylated liposomes (Table 1). We used the ratio of the effect on neutrophil migration and the effect on regeneration or transactivation as a readout for the therapeutic ratio of the different PLP formulations. The results showed that when considering the effect on regeneration, encapsulation in AmbiMACs (20% DSPG) resulted in a ratio (4.707), comparable to that observed for PEGylated liposomes (4.040), and both were considerably higher than the ratio observed for free PLP (0.901). For both types of liposome, this difference with free PLP was due to the fact of both a larger inhibition of the neutrophil migration and a smaller inhibition of the regeneration. When we determined the ratio between the effect on neutrophil migration and the effect on transactivation, a higher ratio was observed for the PEGylated liposomes (3.453) than for AmbiMACs (20% DSPG) (0.603), which was only slightly higher than the ratio observed for free PLP (0.177). This difference between the two types of liposomes was due to the fact that PLP in AmbiMACs (20% DSPG) had a considerably higher effect on transactivation, comparable to free PLP. In conclusion, our zebrafish model was demonstrated to be a useful model for assessing the therapeutic ratio of novel liposomal GC formulations.

## 4. Discussion

In the present study, we used the zebrafish embryo model to investigate liposomal drug targeting in vivo. This model enabled us to determine not only the biodistribution of liposomes in real-time, but also the physiological effects of the drug encapsulated in these liposomes, in this case PLP. Using this model, we investigated two types of liposomal formulations, PEGylated liposomes, which were shown to remain in circulation for long periods of time, and AmbiMACs, which are liposomes with a novel formulation (20% DSPG, 50% DOPC, 30% cholesterol) that were shown to specifically target macrophages. In addition, we showed that both types of liposomes accumulated at sites of inflammation, thereby increasing the anti-inflammatory effects of PLP and reducing the side effects of this GC drug.

Our results from confocal microscopy imaging of zebrafish embryos injected with different liposome formulations illustrate the feasibility to visualize and compare the biodistribution of liposomes in vivo, adding to previous studies on nanoparticles as a drug delivery system in zebrafish [52,53,65,66]. We investigated the biodistribution in 3 dpf embryos of a PEGylated (PEG2000-DSPE, DPPC and cholesterol) formulation, which has widely been studied for delivery of PLP [20,21,25,67]. Our images show that these liposomes were mainly circulating in the vasculature at 2 h after intravenous injection and were hardly taken up by macrophages, in agreement with previous reports [53,65]. For these liposomes, however, we observed association with endothelial cells (ECs) of the PCV, CV, and the CHT, which have been shown to function in zebrafish embryos as the equivalent of the scavenger endothelial cells in mammals [52]. These data are therefore in line with the observed accumulation of this type of liposome in spleen and liver upon intravenous injection in rats [19]. Clearance of liposomes from the circulation by SECs, especially in the liver, is a critical problem for the application of the liposome drug delivery systems [7,15], so we suggest the use of our zebrafish model system to screen altered formulations of the PEGylated liposome for reduced association with SECs in future studies.

In addition to using PEGylated liposomes, we used a novel liposomal formulation, AmbiMAC, which is based on the marketed liposomal product AmBisome. Like other anionic nanoparticles, AmBisome liposomes have been shown to strongly interact with the scavenger receptor Stabilin-2 on SECs and blood-resident macrophages after injection in zebrafish embryos [52]. Upon replacement of the saturated phospholipid DSPC by the unsaturated DOPC, the liposomes (now called AmbiMACs) also showed interaction with tissue-resident macrophages (unpublished data). This effect is probably a result of the decreased rigidity of the newly designed liposomes, enabling them to penetrate the tissue more easily. In the present study, we have optimized the formulation of AmbiMACs by varying the DSPG:cholesterol ratio of a liposome that for the other 50% consisted of DOPC, with the aim to have PLP delivered to sites of inflammation by macrophages recruited to these areas. Our images show that low ratios prevent macrophage uptake and that high ratios induce more targeting to the area that includes CV and CHT, consistent with the finding that these cells preferentially scavenge anionic nanoparticles [52]. The ratio we selected for further studies showed maximal macrophage targeting with lowest association with venous ECs among the formulations tested in this study. It should be pointed out that the macrophage targeting of the AmbiMACs will probably complicate their clinical application, because the vast majority of these liposomes will most likely be cleared by Kupffer cells in the liver upon systemic administration. Therefore, it will be interesting to use local administration of AmbiMACS into the inflamed tissue in future studies.

In order to test these liposomes for the delivery of PLP to inflamed tissues, we used laser wounding in the tail of the zebrafish embryo. In previous studies we used tail fin amputation to study anti-inflammatory effects of GCs in zebrafish [49,50], but for the present study wounding in a better vascularized part of the body seemed a more relevant model system. Both types of tested liposomes accumulated in the wounded area. The PEGylated liposomes showed a diffuse distribution at the wounded site, indicating extravasation of the liposomes in this specific area, whereas the AmbiMACs were localized inside the macrophages that had migrated towards the wound. Interestingly, PLP encapsulated in either liposome type was more potent in inhibiting the wounding-induced neutrophil migration compared to free PLP. Thus, using our zebrafish model, we demonstrated an enhanced anti-inflammatory effect of PLP upon encapsulation with a new macrophage-targeting liposome formulation. The inhibition of neutrophil recruitment is unlikely to be a direct result of PLP action on macrophages, since we have observed that macrophage ablation does not affect the GC-induced inhibition of neutrophil migration (unpublished observation) and that the decrease in the expression of neutrophil-specific chemokines in the inflamed tissue appears to be an important mechanism underlying the GC effect on the recruitment of these cells [49]. We therefore assume that macrophage-targeting results in increased local exposure of the inflamed tissue to PLP, in addition to the direct anti-inflammatory effects on macrophages. Furthermore, encapsulation in PEGylated liposomes was shown to enhance the anti-inflammatory effect of PLP as well, similarly to the enhancement observed for this type of liposomes in mammalian models [16,19,20,21,22,67], which validates our zebrafish model. The PEGylated liposome-encapsulated PLP was slightly less potent in suppressing the neutrophil migration than the PLP encapsulated in AmbiMACs, which suggests a higher delivery efficiency of liposomes to the inflamed site through macrophage accumulation than through passive accumulation.

In order to improve GC therapies, an important aspect that should be taken into consideration is the severity of the side effects [3]. Therefore, in our study we used two in vivo assays to assess possible adverse effects of PLP. First, we determined the inhibition of regeneration of the tail upon fin amputation, since inhibited wound healing or tissue regeneration is a commonly observed side effect of GC treatment [68]. It could be argued that GC effects on tissue regeneration result from the inhibition of the inflammatory response, which has been shown to be required for the regenerative process [69]. The regeneration of the tail fin in zebrafish larvae has been shown to be dependent on the controlled recruitment and action of immune cells, in particular of macrophages producing pro-inflammatory cytokines, such as TNF-α [70], have previously shown that GC treatment of zebrafish larvae inhibits the differentiation of macrophages to a pro-inflammatory phenotype [49], the anti-inflammatory effect of GCs, and their inhibitory effect of regeneration could be considered as interconnected. However, in a recent study on ginsenosides, a specific class of GR agonists, we have demonstrated that the anti-inflammatory effects of Gr activation are not necessarily associated with an inhibition of tissue regeneration upon tail fin amputation [71]. These results indicate that it is possible to develop anti-inflammatory GC drugs which leave the regenerative capacity of the damaged tissue intact and show that our animal model enables studying regeneration as a side effect of GC treatment independent of the immune-suppressive effects. Second, to study the exposure to GCs and activation of GRs outside the inflamed tissue, we used a zebrafish reporter line in which the expression of the GFP gene was driven by a GRE-containing promoter and we determined the GFP expression throughout the body of the embryos upon administration of free PLP and encapsulated PLP [58].

The findings from both assays showed that encapsulation in either type of liposome reduced the effects of PLP, suggesting that liposome encapsulation may increase the therapeutic ratio of PLP not only by enhancing the desired anti-inflammatory effects, but also by decreasing the adverse effects. Interestingly, in the tail fin regeneration assay both liposomes showed similar effects, which were decreased compared to the effect of free PLP, but in the GRE:GFP reporter line the AmbiMACs showed a a much higher induction of the GFP expression (almost comparable to the effect of free PLP) than the PEGylated liposomes. This difference may be related to the different pharmacokinetics of the liposomes, since the effect of GCs in the tail fin regeneration assay depends on their presence during the first hours after amputation [62,71], whereas the observed effect on GFP expression results from their activity over almost the entire time between the injection and the imaging (24 h).

In conclusion, we present the zebrafish model system as a useful tool for studies on liposomal encapsulation of GCs. Exploiting the transparency of this model, we were able to image the biodistribution of PEGylated liposomes and novel macrophage-targeting liposomes, and both liposomes showed accumulation at sites of inflammation. Encapsulation of PLP in these liposomes enhanced the anti-inflammatory effects of this drug and reduced its adverse effects. We suggest the use of this model for future preclinical studies aimed at the optimization of liposomal formulations of anti-inflammatory GC drugs, and that both PEGylated liposomes and AmbiMACs form interesting starting points for these studies.

## Figures and Tables

**Figure 1 cells-11-00671-f001:**
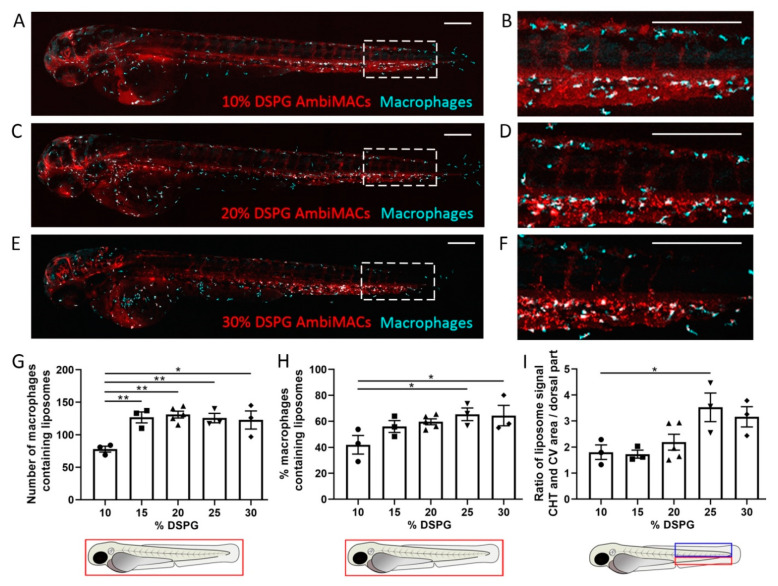
Biodistribution of AmbiMACs with different formulations in zebrafish embryos. (**A**–**F**) Representative images of *Tg(mpeg1:GFP)* embryos injected with AmbiMACs containing different percentages of DSPG at 2 days post-fertilization (dpf). Confocal microscopy images were taken at 2 h post-injection (hpi). AmbiMACs are shown in red and macrophages in cyan. The tail regions (indicated by the dashed boxes in (**A**,**C**,**E**)) are shown at higher magnification in (**B**,**D**,**F**). Scale bar = 200 μm. (**G**,**H**) The number (**G**) and percentage (**H**) of macrophages containing AmbiMACs quantified in the whole body. A significant difference was observed for the number of macrophages containing AmbiMACs with DSPG percentages of 15–30% compared to 10%. For the percentage of macrophages containing liposomes, a significant difference was observed for AmbiMACs with 25% and 30% DSPG compared to the 10% DSPG. (**I**) The ratio between the (fluorescent) signal of AmbiMACs in the area, indicated by the red box, encompassing the caudal vein (CV) and the caudal hematopoietic tissue (CHT), and the signal in the dorsal part of the tail (indicated by the blue box). A significant difference was observed between injection with AmbiMACs (25% DSPG) compared to AmbiMACs (10% DSPG). Statistical analysis was performed by one-way ANOVA with Bonferroni’s post hoc test. Data shown are the mean ± SEM of 3–5 individual embryos, of which the individual data are indicated. Statistically significant differences between groups are indicated by: * *p* < 0.05; ** *p* < 0.01.

**Figure 2 cells-11-00671-f002:**
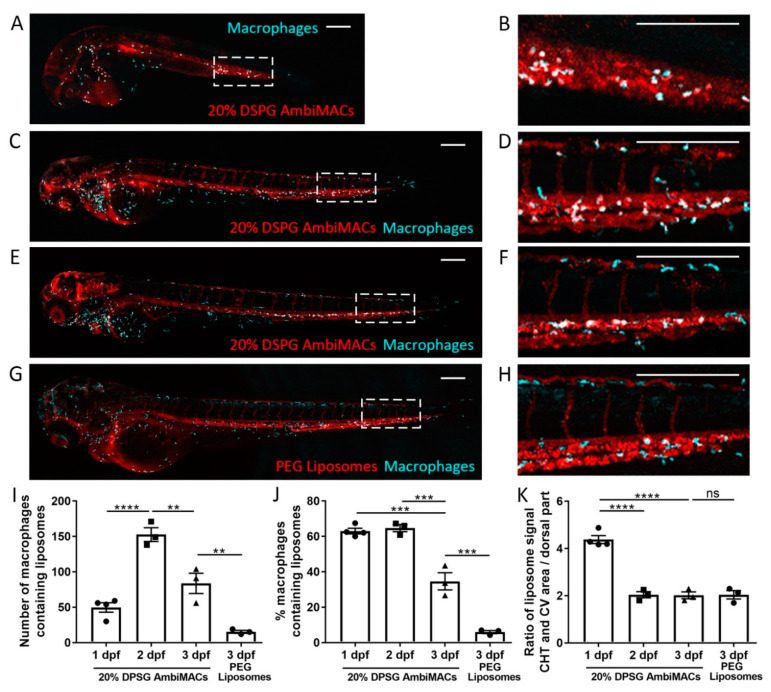
Biodistribution of liposomes in zebrafish embryos at different stages of development. (**A**–**H**) Representative images of embryos of the *Tg(mpeg1:GFP)* line injected with AmbiMACs (20% DSPG) at 1 (**A**,**B**), 2 (**C**,**D**), or 3 dpf (**E**,**F**) or with PEGylated liposomes at 3 dpf (**G**,**H**). Confocal microscopy images were taken at 2 hpi. Liposomes are shown in red and macrophages in cyan. The tail regions (indicated by dashed boxes in (**A**,**C**,**E**,**G**)) are shown at higher magnification in (**B**,**D**,**F**,**H**). Scale bar = 200 μm. (**I**,**J**). The number (**I**) and percentage (**J**) of macrophages containing liposomes quantified in the whole body of embryos injected with AmbiMACs (20% DSPG) at 1, 2, or 3 dpf or with PEGylated liposomes at 3 dpf. Statistical analysis by one-way ANOVA showed a significantly lower number and percentage of macrophages containing liposomes when injected with PEGylated liposomes at 3 dpf, compared to injection with AmbiMACs (20% DSPG). Upon injection with AmbiMACs (20% DSPG), at 1 and 3 dpf, significantly lower numbers of macrophages containing liposomes were observed than at 2 dpf. Embryos at 3 dpf showed significantly lower percentages of macrophages containing liposomes compared to embryos at 1 and 2 dpf. (**K**) The ratio between the (fluorescent) signal of liposomes in the CHT/CV area and the dorsal part of the tail (as described in Figure 1I). The ratio at 1 dpf was significantly higher compared to 2 and 3 dpf. No significant difference was observed between the AmbiMACs (20% DSPG) and PEG liposomes at 3 dpf. Statistical analysis was performed by one-way ANOVA with Bonferroni’s post hoc test. Data shown are the mean ± SEM of 3–4 individual embryos of which the individual data are indicated. Statistically significant differences between groups are indicated by: ** *p* < 0.01; *** *p* < 0.001; **** *p* < 0.0001.

**Figure 3 cells-11-00671-f003:**
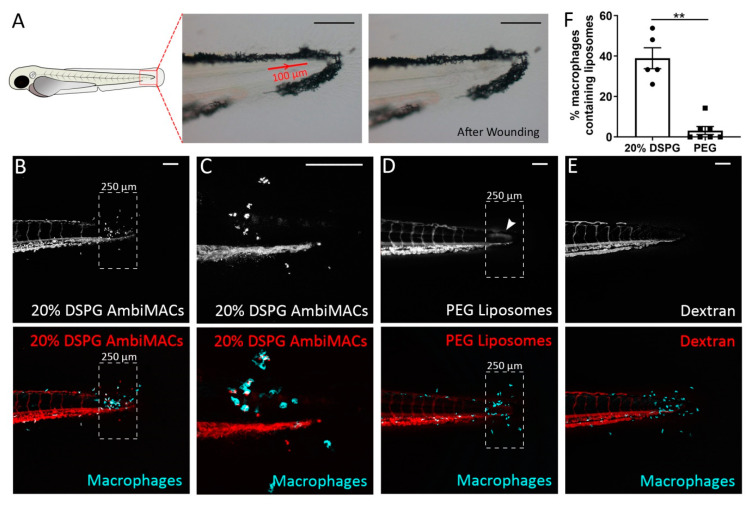
Laser wounding in zebrafish embryos and the subsequent accumulation of liposomes at the wounded site. (**A**) Schematic drawing of a zebrafish embryo at 3 dpf, and brightfield microscopy images showing the position and size of the area exposed to laser irradiation (red line with arrow indicating the direction of laser) and the damaged tissue after the laser wounding procedure. (**B**–**E**) Representative confocal microscopy images of tail regions from 3 dpf embryos of the *Tg(mpeg1:GFP)* line subjected to laser wounding and injected with AmbiMACs (20% DSPG) (**B**,**C**), with (**C**) at higher magnification), PEGylated liposomes (**D**), or dextran (2,000,000 MW, (**E**)). The dashed boxes indicate the area where damage and accumulation of neutrophils were seen. The white arrowhead indicates the accumulation of PEG liposomes (**D**). Images were taken at 4 h post-wounding (hpw). The dashed box shows the area of quantification. Liposomes are shown in red and macrophages in cyan. Scale bar = 100 μm. (**F**). The percentage of macrophages containing liposomes in the area near the laser wound (dashed box), in embryos injected with AmbiMACs (20% DSPG) or PEGylated liposomes. A significantly higher percentage of macrophages containing liposomes was observed upon injection with AmbiMACs (20% DSPG). Statistical analysis was performed by two-tailed *t*-test. Data shown are the mean ± SEM of 5–7 individual embryos, of which the individual data are indicated. Statistically significant differences between groups are indicated by: ** *p* < 0.01.

**Figure 4 cells-11-00671-f004:**
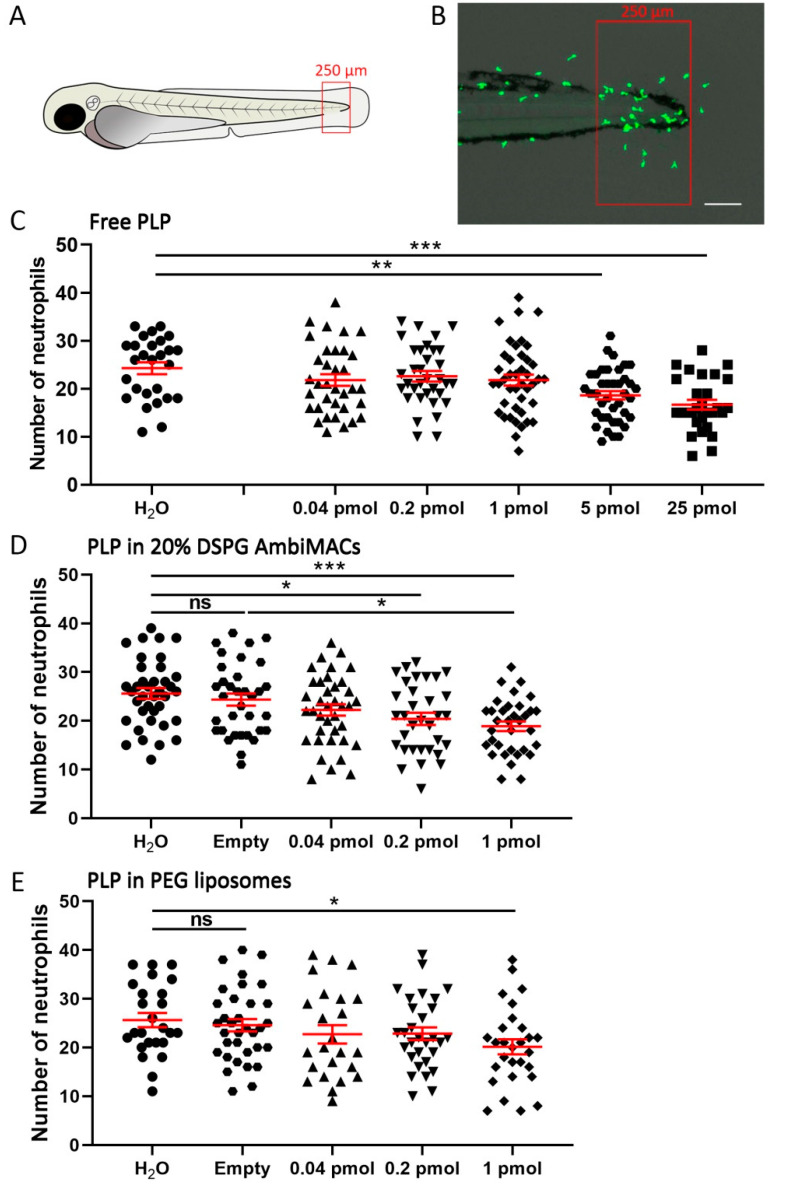
Liposome encapsulation of PLP increased its inhibitory effect on neutrophil recruitment upon laser wounding. *Tg*(*mpx:GFP*) embryos (at 3 dpf) were subjected to the laser wounding procedure and injected with different doses of free or liposome-encapsulated PLP, and fluorescence microscopy images were taken at 4 hpw. (**A**) Schematic drawing of a zebrafish embryo at 3 dpf indicating the area in which the recruited neutrophils were counted (red box). (**B**) Representative image showing the accumulation of neutrophils (green) near the wound. Scale bar = 100 μm. (**C**–**E**) The number of neutrophils recruited to the wounded area at 4 hpw are shown after injection of different doses of free PLP (**C**), PLP encapsulated in AmbiMACs (20% DSPG) (**D**), and PLP encapsulated in PEGylated liposomes (**E**); H_2_O and empty liposomes were injected as control. Statistical analysis was performed by one-way ANOVA with Bonferroni’s post hoc test. A significant inhibition of the neutrophil migration was observed when embryos had been injected with 5 or 25 pmol of free PLP, 0.2 or 1 pmol of PLP encapsulated in AmbiMACs (20% DSPG) and 1 pmol of PLP encapsulated in PEGylated liposomes. Each data point represents a single embryo, and the means ± SEM of data accumulated from three independent experiments are shown in red. Statistically significant differences between groups are indicated by: ns, non-significant; * *p* < 0.05; ** *p* < 0.01; *** *p* < 0.001.

**Figure 5 cells-11-00671-f005:**
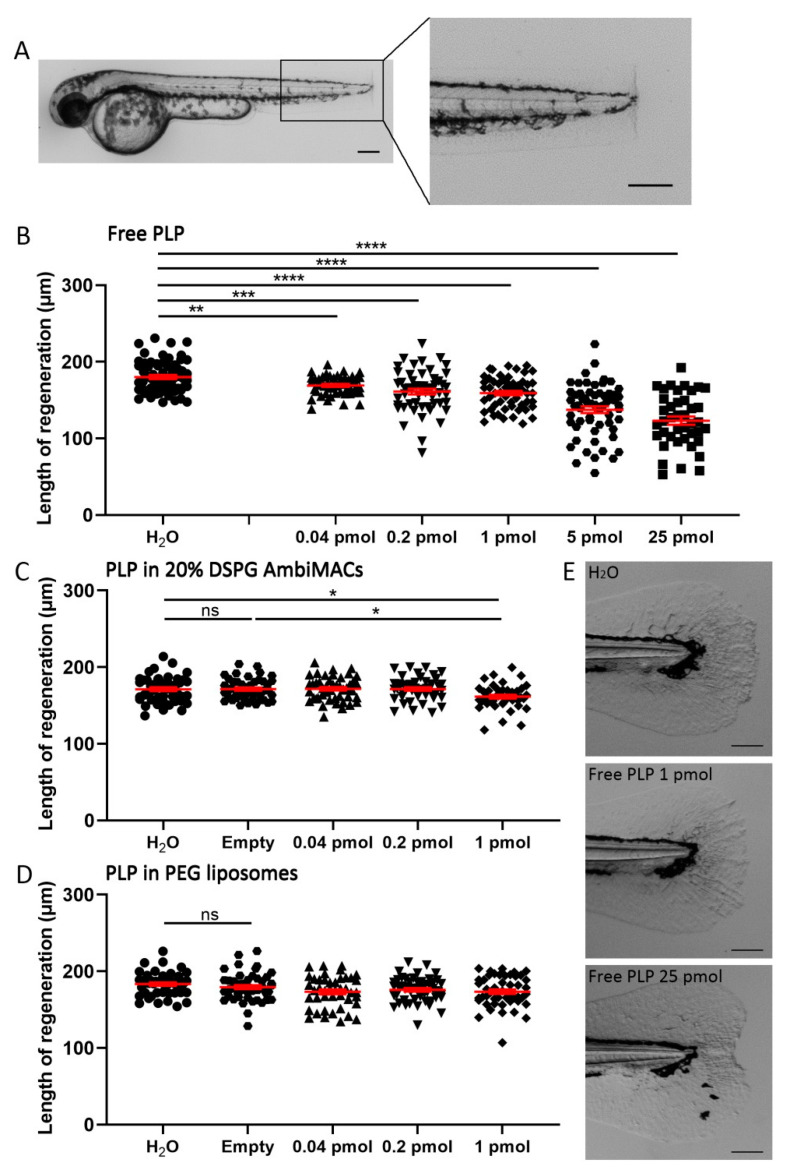
Liposome encapsulation of PLP decreases its inhibitory effect on regeneration of the tail fin after amputation. Embryos (at 2 dpf) were subjected to the tail fin amputation procedure and injected with different doses of free or liposome-encapsulated PLP. (**A**) Representative image of a 2 dpf zebrafish embryo immediately after amputation, showing the position of the amputated part of the tail fin. Scale bar = 200 μm. (**B**–**D**) The length of the regenerated tail fin, measured at 36 h post-amputation (hpa), are shown after injection of different doses of free PLP (**B**), PLP encapsulated in AmbiMACs (20% DSPG) (**C**) and PLP encapsulated in PEGylated liposomes (**D**); H_2_O and empty liposomes were injected as control. Statistical analysis was performed by one-way ANOVA with Bonferroni’s post hoc test. Significant inhibition of the tail fin regeneration was observed when embryos had been injected with 0.04–25 pmol of free PLP or 1 pmol of PLP encapsulated in AmbiMACs (20% DSPG). No significant inhibition was observed after injection with PLP encapsulated in PEGylated liposomes. Each data point represents a single embryo, and the means ± SEM of the data accumulated from three independent experiments are shown in red. Statistically significant differences between groups are indicated by: ns, non-significant; * *p* < 0.05; ** *p* < 0.01; *** *p* < 0.001; **** *p* < 0.0001. (**E**) Representative images of regenerated tail fins at 36 hpa for embryos injected with H_2_O, 1 pmol PLP, or 25 pmol PLP. Scale bar = 100 μm.

**Figure 6 cells-11-00671-f006:**
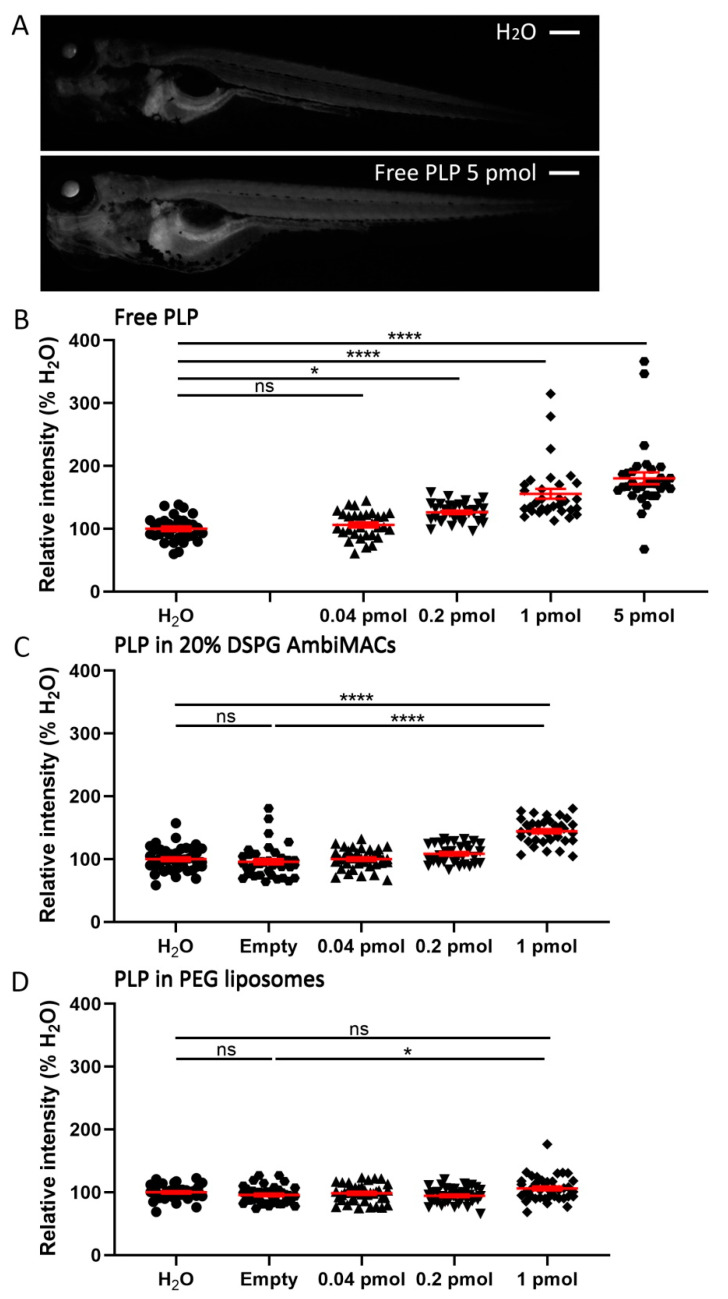
Liposome encapsulation of PLP decreased its effect on the systemic Gr transactivation activity. Embryos (3 dpf) of the *Tg(GRE:GFP)* line were injected with free or liposome-encapsulated PLP, and fluorescence microscopy images were taken at 24 hpi. (**A**) Representative images of *Tg(GRE:GFP)* embryos injected with H_2_O or 5 pmol PLP, showing the GFP signal, which is a readout for the transactivation activity of Gr, which increased after PLP injection. Scale bar = 200 μm. (**B**–**C**) The quantified GFP signals in the *Tg(GRE:GFP)* embryos at 24 hpi are shown after injection of different doses of free PLP (**B**), PLP encapsulated in AmbiMACs (20% DSPG) (**C**), and PLP encapsulated in PEGylated liposomes (**D**); H_2_O and empty liposome were injected as control. Statistical analysis was performed by one-way ANOVA with Bonferroni’s post hoc test. Significant increases in the GFP signal were observed when embryos were injected with 0.2–5 pmol free PLP and 1 pmol of PLP encapsulated in either AmbiMACs (20% DSPG) or in PEGylated liposomes. Each data point represents a single embryo, and the means ± SEM of the data accumulated from three independent experiments are shown in red. Statistically significant differences between groups are indicated by: ns, non-significant; * *p* < 0.05; **** *p* < 0.0001.

**Table 1 cells-11-00671-t001:** Ratio between the therapeutic effect and side effects of 1 pmol PLP treatment.

**Treatment**	$\frac{{\mathrm{Inhibition}\;\mathrm{of}\;\mathrm{neutrophil}\;\mathrm{migration}\;(\%)\;}^{1}}{\mathrm{Inhibition}\;\mathrm{of}\;\mathrm{regeneration}\;(\%)}$		$\frac{\mathrm{Inhibition}\;\mathrm{of}\;\mathrm{neutrophil}\;\mathrm{migration}\;(\%)}{\mathrm{Increase}\;\mathrm{in}\;\mathrm{transactivation}\;(\%)}$	
**Free PLP**	0.901	(9.8/10.9)	0.177	(9.8/55.6)
**PLP in AmbiMACs (20% DSPG)**	4.707	(26.8/5.7)	0.603	(26.8/44.5)
**PLP in PEGylated liposome**	4.040	(21.4/5.3)	3.453	(21.4/6.2)

^1^ All effects are determined as the percentage difference relative to the H_2_O treatment (values are also shown in Figure 4, Figure 5 and Figure 6).

## Data Availability

Data sharing is not applicable to this article.

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
