# Peer review of "Two Types of Liposomal Formulations Improve the Therapeutic Ratio of Prednisolone Phosphate in a Zebrafish Model for Inflammation"

_cells, 2022, doi:10.3390/cells11040671_

Round 1
Reviewer 1 Report
In their manuscript “Liposome encapsulation of prednisolone phosphate improves its therapeutic ratio in a zebrafish model for inflammation”, the authors examine the effects of injecting the glucocorticoid prednisolone phosphate into zebrafish embryos either directly or encapsulated into two different liposome preparations, one of which specifically targets macrophages. They observe that liposome mediated delivery reduces neutrophil migration to a laser-induced wound, an inflammation-related process, at lower drug concentrations than injection of the free drug. Furthermore, liposome mediated delivery was less efficient when examining two model readouts for drug side effects. In one of the readouts, reporter gene expression, the effect of the macrophage-targeted liposome preparation was only slightly lower than that of the free drug, showing differential effects of the two ways of encapsulation.
The manuscript addresses an important topic in glucocorticoid drug therapy, improved targeting of the therapeutic effects and reduction of side effects. It nicely illustrates how the zebrafish model can help to examine such topics in vivo. Even though some of the observed effects appear of minor magnitude, they are statistically significant, suggesting that the assays can be used for further improvement of liposome-mediated drug delivery. The manuscript is likely to be of interest for the “Cells” readership.
Overall design and conduction of the experiments appear sound. However, I have a few comments on the description of the assays and on the discussion of the results:
- Please indicate in the materials and methods section how the liposome preparations were fluorescently labelled for imaging, and which excitation/emission wavelengths were employed for imaging. (At the moment, information on the type of dye used [Lissamine Rhodamine PE] is found only in the legend of Suppl. Fig. S4).
- Please describe in more detail how analysis of the co-localization of the liposome and macrophage markers was conducted. Were single z-stacks examined? From the images shown in Figs. 1-3, it is generally very difficult to see which macrophages have taken up the liposome dye. Could one include higher resolution images similar to panel C of Fig. 3 for all the conditions examined (or at least show them as a supplementary figure)?
- Tail fin regeneration is examined as a model for drug side effects, with glucocorticoids inhibiting regeneration. However, tail fin regeneration is dependent on the action of immune cells and requires their migration to the site of amputation. As migration of immune cells is taken as a readout for the desired anti-inflammatory effects of the tested preparations, it appears that both “therapeutic” and “side effect” readouts are interconnected. Could the authors elaborate more on this aspect of their assays in the discussion section?
- The authors use a fluorescent reporter line for glucocorticoid signaling as an additional side effect readout. Besides the differences in global fluorescence induction reported in the manuscript, did the authors observe any spatial differences in the signal with the different preparations? Does the macrophage-targeting preparation lead to increased reporter gene induction in this cell type?
- Could one use the line together with the wounding/regeneration protocols to examine if indeed “PLP [is] delivered to sites of inflammation by macrophages recruited to these areas” (line 539f), by testing whether reporter gene activation in these areas is increased with respect to the global increase reported in the manuscript? Could the effect of glucocorticoids on neutrophil migration also be mediated via effects of the glucocorticoid-liposome uptake on the macrophages themselves and their behavior/signals? Maybe such questions could be further explored in the discussion section.
Author Response
- The reviewer asked us to provide information on the fluorescent label to the Materials and Methods section. We have added this information to this section (Lines 139-141 and 208-209).
- The reviewer asked for a description of how we studied the colocalization of macrophages and liposomes. This description has been added to the Materials and Methods section (Lines 230-241). In addition, the reviewer asked for clearer images of this colocalization. We have added higher magnification images to Figures 1 and 2 which should facilitate the evaluation of the macrophage/liposome colocalization.
- The reviewer points out that inflammation and regeneration are linked processes, which may complicate making a distinction between therapeutic and adverse effects in our model. It has indeed been shown that the inflammatory response, in particular the recruitment and action of macrophages, is required for tissue regeneration upon injury. However, in a recent article we have demonstrated that a certain class of glucocorticoid receptor agonists inhibit the inflammatory response after tail fin amputation in zebrafish embryos, but that these compounds leave the regeneration of the tail fin unaffected. These results show that it is possible to develop glucocorticoid drugs with anti-inflammatory action which leave the regeneration of damaged tissue intact and that we can investigate the glucocorticoid inhibition of the inflammatory and the regenerative response to tissue damage separately in our model. We have added a paragraph on this issue to the Discussion (Lines 621-636).
- The reviewer notes that it would be interesting to show a more detailed analysis of the images of the GRE:GFP reporter line, to show spatial differences and a possible increase in macrophages and/or in the inflamed area. Although this is an interesting question, we would like to point out that the resolution of the images that can be obtained using this reporter line is relatively low, which makes it very difficult to obtain quantitative data on for example organ-specific GFP signals, let alone increased signals in immune cell types that are distributed as individual cells through the body. In addition, to be able to properly identify organs and/or cell types, this would require crossing the line with another transgenic line in which the relevant cell type is labeled. Therefore, we have limited the quantitative analysis of these images to the integrated fluorescence intensity, as in all other published articles in which this line has been used that we are aware of. Furthermore, the reviewer asks whether the effect on neutrophil migration may be a secondary effect of the macrophage targeting. This is unlikely and we have added a brief comment to the Discussion, in which we explain this based on previous data (Lines 602-610).
Reviewer 2 Report
This is a well-organized and well-illustrated paper, has an important clinical message, and should be of great interest to the readers. This research article focused on the development of liposomal formulation of prednisolone phosphate for anti-inflammatory applications in zebra fish. Paragraphing is concise and good, and the article consists of major recent advancements in the field of liposomal drug delivery systems for anti-inflammatory drug delivery and deserves publication after some revisions.
- Please briefly explain the mechanism through which the NP’s can be retained specifically in macrophages in the discussion section?
- Did the authors check the drug loading percentage? If yes, please mention the loading percentage?
- Did the authors check drug release kinetics of the proposed formulation?
- Why did the authors not check the free drug biodistribution or the accumulation in macrophages?
- Without comparison with the free drug, how can the authors claim the advantages of nanoparticles related to toxicity, biodistribution?
Author Response
- The reviewer asked for an explanation of the AmbiMAC macrophage-targeting in the Discussion. We have added a brief explanation in the Discussion (Lines 570-578), since this process was already (more elaborately) explained in the Results section (Lines 260-275).
- As requested by the reviewer, we have added drug loading percentages to Supplementary Table 1. An explanation of how these percentages were calculated has been added to the Materials and Methods section (Lines 163-175).
- The reviewer asked if we checked the drug release kinetics of our liposomal formulations. We did not study these kinetics, because the focus of our manuscript was on targeting and biodistribution of the liposomes, and not the pharmacokinetics/-dynamics. However, in a future study we will certainly study kinetic parameters as well, since they are obviously important characteristics for novel drug formulations.
- The reviewer wonders why we did not check the biodistribution and/or accumulation in macrophages of the free drug. Although this would be interesting and highly relevant to study, a functional fluorescently labeled glucocorticoid drug is not available. A fluorescein-labeled dexamethasone is commercially available, but the label interferes with the action of the steroid in our experiments, and changes the size, lipophilicity and charge of the drug which will most likely affect its biodistribution upon liposomal encapsulation. Therefore, we used a non-labeled glucocorticoid and used the effects of this steroid as a readout.
- The reviewer would like to know how we can evaluate the advantage of our liposomal formulations related to toxicity and biodistribution without using the free drug as a control. As stated in point 4, the biodistribution of the free drug is very difficult to determine, and we therefore studied the biodistribution of the different liposomal formulations. Subsequently, we studied the therapeutic and adverse effects of the liposomal formulations and we did use the free drug as a control, so the advantages of the liposomal formulation of the drug could be evaluated at this functional level.
Reviewer 3 Report
In this manuscript, Y. Xie et al. have used zebrafish embryos to study the biodistribution of liposomes and to determine the therapeutic and adverse effects of prednisolone phosphate encapsulated in these formulations. The authors state that PEGylated liposomes remain in circulation for long periods of time, and AmbiMACs liposomes selectively targeted macrophages., whereas both types of liposomes accumulate near the wounding site favouring the inhibition of the inflammatory response.
It was already demonstrated in rodent models that encapsulated prednisolone in PEGylated liposomes have longer plasma half-life and accumulate in the inflamed joints (Ref. 19,21). In view that similar results were also obtained reported in models for inflammation-related diseases such as atherosclerosis [22], multiple sclerosis [16] and cancer [23], the obvious question is: what is new in the present study? Moreover, all the other studies were achieved in a rodent model, whereas in this study the authors use a zebrafish model. What is the relative advantage of repeating already known results if the experimental evidence and conclusions are the same as those obtained in a mammalian system and the final aim of these studies is the clinical practice?
Critical information is also missing. This reviewer has had a very hard time trying to understand what the authors wished to explain. Just as a simple example: the authors claim that AmbiMAC liposomes are selectively targeted to macrophages. When the reader reaches the end of the article, the most specific information was provided in the figure legends: “macrophages are shown in cyan” (?). First sentence of Results (lines 229-230). “A new liposomal formulation, that had previously been shown to target macrophages…” Where was it shown? Regardless of this, this implies that even this observation is not novel either.
The authors’ conclusions are (literal): 1) further optimization of liposomal formulations of GCs represents an interesting research avenue, and 2) zebrafish embryos are a useful tool for investigation. No comments.
Author Response
The reviewer mainly questions the novelty and relevance of our results. Apparently, we did not explain the novelty of our data and the added value of the zebrafish model well enough, especially in the conclusions. Based on the reviewer’s criticism, we have rewritten large parts of the manuscript (Title, Abstract (Lines 25-28), Introduction (Lines 117-124) and Discussion (Lines 543-552 and 653-661)).
In these rewritten sections, we have attempted to point out more clearly that:
- We have developed, using zebrafish embryos, a vertebrate animal model that enables us to study the real-time biodistribution of liposomes (containing glucocorticoid drugs) by direct visualization, in combination with readouts of both the therapeutic and adverse effects of the glucocorticoids. Such a study would be impossible in any mammalian model, and given the possibilities to use this model for higher throughput studies, this model is highly suitable for future studies on the screening and optimization of novel liposomal formulations of glucocorticoids.
- We have used our model to study two types of liposomal formulations. First, PEGylated liposomes were used, since they have been shown to increase the therapeutic ratio of glucocorticoid drugs in mammalian systems, so they represent a logical starting point and can be used for validation of the system. Interestingly, although the PEGylated liposomes increased the therapeutic ratio, the images of their distribution in the embryonic bodies showed that there is significant interaction with the RES, which shows that the targeting of these liposomes can still be improved. Second, we used AmbiMACs, a novel (i.e. not published before) liposomal formulation which uniquely targets both blood- and tissue-resident macrophages. These liposomes also enhanced the therapeutic ratio of the glucocorticoids, which demonstrates that macrophage targeting is also an interesting approach for the development of novel liposomal formulations of glucocorticoids.
Round 2
Reviewer 3 Report
I still think that the article is not adding any new piece of information regarding what is already known (including those previous publications by the authors). Unfortunately, new texts do not add too much to the concepts of the original version. On the other hand, they do clarify some missing technical issues, which was extremely necessary.